# Factors related to met needs for rehabilitation 6 years after stroke

**Charlotte Ytterberg**[1,2,3]*, **Hanne Kaae Kristensen**[3,4], **Malin Tistad**[1,5], **Lena von Koch**[1,2]

**1** Department of Neurobiology, Care Sciences and Society, Karolinska Institutet, Huddinge, Sweden, **2** Karolinska University Hospital, Huddinge, Sweden, **3** Research Unit in Rehabilitation, Institute of Clinical Research, University of Southern Denmark, Odense, Denmark, **4** Health Sciences Research Centre, University College Lillebaelt, Odense, Denmark, **5** School of Education, Health and Social Studies, Dalarna University, Falun, Sweden

* charlotte.ytterberg@ki.se

## Abstract

### Introduction

Research on stroke rehabilitation mainly concerns the first year of recovery, and there is a lack of knowledge regarding long-term rehabilitation needs and associated factors.

### Aim

The aim was to explore the perceived needs for rehabilitation services of people six years after stroke and factors associated with having rehabilitation services needs met.

### Methods

The study was a 6-year follow up of a prospective study on the rehabilitation process after stroke. Data on perceived needs for rehabilitation, personal factors, disease specific factors, and patient-reported disability were collected through face-to-face interviews in the partici-pants' homes. Logistic regression models were created to explore associations between having rehabilitation services needs met in 11 problem areas (dependent variable) and the independent variables: involvement in decisions regarding care and treatment, sex, age, sense of coherence, self-defined level of private financing, stroke severity, frequency of social everyday activities, perceived impact of stroke, and life satisfaction.

### Results

The 121 participants had a mean age of 63 years at stroke onset and 58% were men. In all problem areas the majority (53–88%) reported having needs met at six years after stroke, however 47% reported unmet needs regarding fatigue and 45% regarding mobility. A lower perceived impact on participation was found to be associated with having rehabilitation services needs met in seven problem areas: mobility, falls, pain, fatigue, concentration, memory, and sight. The strongest association for having needs met was found for the independent variable, involvement in care and treatment, within the three problem areas mobility, falls, and speaking.

**Data Availability Statement:** Since data can indirectly be traced back to the study participants, according to the Swedish and EU personal data sharing legislation, access can only be granted upon request. Request for access to the data can

be put to our Research Data Office (rdo@ki.se) at Karolinska Institutet, and will be handled according to the relevant legislation. In most cases, this will require a data processing agreement or similar with the recipient of the data.

**Funding:** LvK: Swedish Research Council 2007-3087 and 2013-2806 http://www.vr.se/, Stockholms Läns Landsting 2006-0700 https://www.sll.se/. CY: Swedish Stroke Association www.strokeforbundet.se/ and the Promobilia Foundation. The funders had no role in study design, data collection and analysis, decision to publish, or preparation of the manuscript.

**Competing interests:** The authors have declared that no competing interests exist.

## Conclusion

In a long-term perspective, there were several modifiable factors associated with having rehabilitation services needs met. The most prominent were perceived involvement in care and treatment, and perceived participation. These factors had a stronger association with having rehabilitation services needs met than disease specific factors six years after stroke.

## Introduction

Stroke is a major health problem due to high prevalence, long-lasting disability [1] and social impact [2]. Even though progress has been made in diagnostics and acute treatment, a stroke often has a significant negative impact on a person's health and frequently leads to a wide range of activity limitations and participation restrictions, even in a long-term perspective [3, 4]. Hence, active rehabilitation early after a stroke is recommended [5, 6]. It is recommended that rehabilitation after stroke should include a holistic perspective, the active involvement of people with stroke, and the services of a multidisciplinary team [6]. The team should have professional knowledge, skills and experience to work in partnership with people with stroke and their close relations [7]. However, rehabilitation interventions and research in rehabilitation after stroke often focus on the first year of recovery, and there is less knowledge on functioning in everyday life and the needs for rehabilitation and support in the long-term. Nevertheless, studies show that there are persistent long-term consequences after stroke [8–10], indicating a need to increase the understanding of how such consequences can be reduced. Furthermore, many people, with persisting declined functioning post stroke, report unmet needs for rehabilitation in many different areas [11–16].

Factors that have been reported to be associated with unmet needs for rehabilitation after stroke constitute higher [14] or lower age at stroke onset [15, 17], a more severe stroke [11], higher perceived impact of stroke [11], pain [14], depression [14, 15, 17], fatigue [17] dependency in activities of daily living (ADL) [14, 15], greater disability [18–20] and not returning to work [7]. Met needs have been reported to be associated with shared decision-making in goal-setting during rehabilitation [12, 21], number of rehabilitation contacts during the first 4 months [20], and contact with rehabilitation throughout the first year after stroke [22]. However, only a few of the studies mentioned above have a longer perspective than one year after stroke.

Previous studies on factors associated with unmet needs after stroke mainly focus on disease-related factors, impairments and activity limitations as assessed by health professionals. However, as needs identified by health professionals have been reported not to capture all those identified by people with stroke [23], the individual's perspective should also be included as recommended by WHO [24]. In partnership with the healthcare professionals, the patients with stroke should be actively involved in the rehabilitation process, be able to express their needs and values, and have the opportunity to make informed decisions about their rehabilitation [21, 25–28]. Involving patients in shared decision-making has been reported to increase patient satisfaction and motivation, and create a greater sense of ownership [12, 29–32]. Thus, when exploring factors underlying the many unmet needs after stroke, not only disease-specific factors and assessments performed by health professionals, but also the patients' perspectives, should be included.

Therefore, the aim of the present study was to explore the perceived needs for rehabilitation services of people six years after stroke, as well as factors (personal, disease specific, and patient-reported) associated with having rehabilitation services needs met.

## Methods

### Participants and procedure

This study was a 6-year follow up of the study Life After Stroke phase 1 (LAS-1), a prospective study of the rehabilitation process during one year after stroke described in detail previously [11, 22, 33]. Originally, 349 patients diagnosed with stroke were consecutively recruited from the stroke units at Karolinska University Hospital between 2006 and 2007. For the 6-year follow-up, participants from LAS-1 who were still alive were contacted, informed about the study and asked to participate. After written informed consent had been obtained, data were collected in one session through face-to-face interviews in the participant's home by research assistants trained for the purpose (occupational therapists or physiotherapists) with extensive experience in rehabilitation after stroke. The participants in the present study were persons included in LAS-1 who agreed to participate in the six-year follow-up and who answered the Stroke Survivor Needs Survey Questionnaire (SSNSQ) [13].

Ethical permission was applied for and then granted by the Regional Ethics Committee in Stockholm both for the original study and the 6-year follow-up study (applications: 2005/1462-31/3, 2011/1573-32 and 2012/428-32).

### Measurements

**Dependent variable.** To assess the participants' perceived needs after stroke, the SSNSQ [13] was used. The questionnaire consists of 44 closed questions with response categories to assess the level of change or needs in seven domains. The SSNSQ was developed to assess perceived needs after stroke and included questions from validated questionnaires. It was validated in a review process by the King's College London Stroke Research Patients and Family Group (a service user research advisory group) [13]. In the present study, 11 questions within the domain "health after stroke" were used concerning needs for rehabilitation services in the problem areas: mobility, falls, incontinence, pain, fatigue, emotion, concentration, memory, speaking, reading, and sight. Participants were asked to choose from five response categories in relation to support received. For example, 'Since your stroke, have you had enough treatment to help improve your mobility (e.g. walking, moving your legs)?'—1. Yes definitely; 2. Yes to some extent; 3. No, I did not get enough treatment; 4. I did not want treatment; 5. I did not have any mobility difficulties. Answers were categorized into needs met (alternatives 1 and 4–5), and needs met to some extent, or unmet (alternatives 2–3).

**Independent variables.** One additional question from the SSNSQ, domain "health after stroke", about involvement in decisions regarding care and treatment was included. Participants were asked to choose from five response categories in relation to involvement: 'Since your stroke, have you been involved as much as you have wanted to be in decisions about your care and treatment?—1. Yes definitely; 2. Yes to some extent; 3. No, but I would have liked to have been more involved; 4. No, but I did not mind; 5. Don't know/Can't say; 6. I have not had any care or treatment since my stroke. Answers were categorized into involved (alternatives 1 and 4) and involved to some extent or not involved (alternatives 2–3).

Data on age and sex were retrieved from the medical records. To assess sense of coherence (SOC) the 13-item version of the SOC scale was used [34]. The SOC scale is a self-report questionnaire consisting of 13 items rated on a seven-graded Likert scale. The total score ranges from 13 (weak SOC) to 91 (strong SOC). An adult individual's SOC is considered to be relatively stable over time [34]. Self-defined level of private financing (sufficient, just sufficient, insufficient) was collected through interview. In the analyses, the answers were aggregated into sufficient and just sufficient/insufficient). Using the Barthel Index [35], which has shown good

agreement with other stroke severity measures, stroke severity was categorized as recommended: mild (scores 50–100), moderate (scores 15–49) or severe (scores ≤14) [36]. In the analyses, the scores were aggregated into mild and moderate/severe. Data on frequency of social everyday activities were collected using the Frenchay Activities Index (FAI) [37]. The FAI consists of 15 items and the score is based on the frequency with which an activity has been performed during the previous 3 or 6 months. The total score ranges from 0 (inactive) to 45 (very active). To assess perceived impact of stroke, the Stroke Impact Scale (SIS) 3.0 was used [38]. The SIS consists of 8 domains: strength, memory and thinking, emotion, communication, activities in daily life, mobility, hand function, and participation. The SIS is made up of 59 items and scores range from 0 (maximum impact) to 100 (no impact). In addition, perceived recovery after stroke is rated on a visual analogue scale ranging from 0 (no recovery) to 100 (full recovery). Data on life satisfaction was collected using the Life Satisfaction Checklist (LiSat-11) [39]. The LiSat-11 is a self-report questionnaire that assesses life satisfaction with the global item "Life as a whole" and ten domain specific items. Answer alternatives range from 1 (very dissatisfied) to 6 (very satisfied). In the present study, the overall item "Life as a whole" was included and categorized as recommended: not satisfied (alternatives 1 to 4) and satisfied (alternatives 5 and 6) [40].

All data were collected at the six-year follow-up except for data on stroke severity and age which were collected within the first week after stroke onset, and data on SOC which were collected at 12 months post stroke.

## Analyses

To analyse differences between participants with met and unmet needs related to the 11 problem areas, univariable logistic regression analyses were performed. Eleven logistic regression models were created to explore associations between having rehabilitation services needs met in regard to each problem area, respectively (dependent variable), and the independent variables. In all models the independent variables were: involvement in decisions on care and treatment, age, sex, SOC, self-defined level of private financing, frequency of social everyday activities, SIS domain corresponding to the dependent variable or stroke severity in cases where no corresponding SIS domain was identified, SIS participation, SIS recovery, and life satisfaction. Participants with missing data in a model were excluded from that particular model. A stepwise forward selection was used where variables with p≤0.05 were entered and those with p≥0.10 were removed. The Enter method was then used to verify a final model with more patients since several variables with missing data may have been excluded. Significance level was set to 0.05. SAS® System 9.4, SAS Institute Inc., Cary, NC, USA was used for the statistical analyses.

## Results

At the 6-year follow-up 121 participants remained in the study; 166 were deceased, 44 declined to take part and 18 could not be traced. Additionally, 11 had not answered the SSNSQ, thus 110 participants were included in the present study. Their mean age at stroke onset was 63 years, ranging from 24 to 85 years, 64 (58%) were men, and 91 (83%) had a mild stroke severity. The mean age of all 349 participants in the original study group at stroke onset was 72 years, ranging from 24 to 95 years, 188 (54%) were men, and 213 (61%) had a mild stroke severity. The mean age at stroke onset of the 239 participants from the original study group who were deceased or non-responders was 76 years, ranging from 37 to 95 years, 125 (52%) were men, and 123 (51%) had a mild stroke severity. There was no difference between study

participants and non-responders in sex (p = 0.203) but the non-responders were significantly older than the participants (p<0.001).

Table 1 presents the characteristics of participants with met and unmet needs related to the 11 problem areas categorized with respect to the independent variables, and p-values from the univariable analyses. In all problem areas the majority reported having met needs, although 47% reported unmet needs related to fatigue problems and 45% related to mobility problems. Participants who perceived a lower impact on participation and a higher recovery after stroke were more likely to report having rehabilitation services needs met in all problem areas except sight.

Results from the logistic regression analyses are shown in Table 2. A lower perceived impact on participation was found to be associated with having rehabilitation services needs met in seven problem areas: mobility, falls, pain, fatigue, concentration, memory, and sight. However, the strongest association was found for the independent variable, involvement in care and treatment, within the three problem areas mobility, falls, and speaking.

## Discussion

This study shows that although the majority report met needs for rehabilitation services six years after stroke, between 19% and 47% report unmet needs within all problem areas studied. The largest proportions of unmet needs were found within the areas mobility and fatigue. Several factors, which varied between the 11 problem areas, were associated with having rehabilitation services needs met. However, the most prominent factors were lower perceived impact on participation and involvement in care and treatment. Since these factors may be modifiable, the results are highly important and of great clinical significance for rehabilitation after stroke.

Almost half of the participants reported unmet needs in relation to fatigue and mobility. These results are in line with previous studies showing that fatigue and mobility problems are common long after the stroke occurred and impact negatively on many aspects of functioning, in particular participation in everyday life [41–44]. Since rehabilitation targeting mobility can yield beneficial effects [45] even in the chronic phase after stroke [46], our results indicate that there is a need for recurrent long-term rehabilitation. For people with remaining disability after stroke one option to maintain mobility might be to have increased access to and possibilities for physical activity in the community [47]. There is also an urgent need to develop and evaluate interventions to reduce fatigue because the evidence base for the effectiveness of interventions targeting fatigue is limited [48].

Participants with a low perceived impact on participation were more likely to report rehabilitation services met needs in seven of the problem areas. Whilst several studies highlight participation as crucial after stroke and with potential to contribute to essential aspects such as a sense of belonging and purpose, identity, autonomy, independence and confidence [49], less is known about how participation best should be supported. Support and services that include social and leisure activities have been prioritized among persons with stroke about one-year post stroke [50]. Moreover, a meta ethnographic review [49] of social participation suggests that persons with stroke build confidence to participate in activities by learning from both health professionals and other persons with stroke, and by their own trying and practicing. Rehabilitation services have been described as having a too strong biomedical focus [51] compared to the patients' focus on regaining former roles and on psychosocial needs. A strong focus not only on participation in rehabilitation, creation of opportunities for peer-learning, but also access to long-term support to promote participation after stroke, could potentially contribute to the positive perception of participation and reduce unmet long-term needs for rehabilitation [52].

**Table 1. Characteristics of participants and univariable analyses of met and unmet needs by problem area.**

|  | Need met | Need met to some extent or Unmet* | P-value |
|---|---|---|---|
| **Mobility problems, n (%)** | 60 (55) | 49 (45) |  |
| Age, median | 65 | 63 | 0.885 |
| Sex, n |  |  |  |
| Men/Women | 39/21 | 25/24 | 0.142 |
| Sense of coherence, median | 81 | 78 | 0.095 |
| Private financing, n |  |  |  |
| Enough/Just enough or Not enough | 41/19 | 20/29 | **0.005** |
| SIS Mobility, median | 96 | 88 | **<0.001** |
| Frenchay Activities Index, median | 32 | 26 | **0.006** |
| Involvement in care and treatment, n |  |  |  |
| Involved/Involved to some extent or Not involved | 43/12 | 16/24 | **<0.001** |
| SIS Participation, median | 92 | 69 | **<0.001** |
| SIS Recovery, median | 90 | 64 | **<0.001** |
| Life Satisfaction, n |  |  |  |
| Satisfied/Not satisfied | 37/18 | 21/23 | 0.052 |
| **Falls, n (%)** | 76 (70) | 33 (30) |  |
| Age, median (IQR) | 64 | 62 | 0.923 |
| Sex, n |  |  |  |
| Men/Women | 46/30 | 18/15 | 0.561 |
| Sense of coherence, median | 78 | 78 | 0.095 |
| Private financing, n |  |  |  |
| Enough/Just enough or not enough | 48/28 | 13/20 | **0.024** |
| SIS Mobility, median | 96 | 82 | **<0.001** |
| Frenchay Activities Index, median | 32 | 24 | **0.002** |
| Involvement in care and treatment, n |  |  |  |
| Involved/Involved to some extent or Not involved | 48/18 | 11/18 | **0.002** |
| SIS Participation, median | 89 | 63 | **<0.001** |
| SIS Recovery, median | 85 | 60 | **<0.001** |
| Life Satisfaction, n |  |  |  |
| Satisfied/Not satisfied | 45/25 | 13/16 | 0.077 |
| **Incontinence problems, n (%)** | 85 (78) | 24 (22) |  |
| Age, median | 64 | 66 | 0.862 |
| Sex, n |  |  |  |
| Men/Women | 51/34 | 13/11 | 0.609 |
| Sense of coherence, median | 81 | 68 | **0.007** |
| Private financing, n |  |  |  |
| Enough/Just enough or not enough | 53/32 | 8/16 | **0.014** |
| Stroke severity |  |  |  |
| Mild/Moderate or Severe | 78/7 | 13/11 | **<0.001** |
| Frenchay Activities Index, median | 32 | 17 | **<0.001** |
| Involvement in care and treatment, n |  |  |  |
| Involved/Involved to some extent or Not involved | 47/28 | 12/8 | 0.827 |
| SIS Participation, median | 86 | 61 | **<0.001** |
| SIS Recovery, median | 80 | 50 | **<0.001** |
| Life Satisfaction, n |  |  |  |
| Satisfied/Not satisfied | 50/28 | 8/13 | **0.036** |
| **Pain, n (%)** | 88 (81) | 21 (19) |  |

(*Continued*)

**Table 1.** (Continued)

| | Need met | Need met to some extent or Unmet* | *P*-value |
|---|---|---|---|
| Age, median | 65 | 62 | **0.044** |
| Sex, n | | | |
| Men/Women | 55/33 | 8/13 | **0.047** |
| Sense of coherence, median | 81 | 69 | **0.008** |
| Private financing, n | | | |
| Enough/Just enough or not enough | 53/35 | 8/13 | 0.071 |
| Stroke severity, n | | | |
| Mild/Moderate or Severe | 76/12 | 15/0 | 0.106 |
| Frenchay Activities Index, median | 30 | 31 | 0.514 |
| Involvement in care and treatment, n | | | |
| Involved/Involved to some extent or Not involved | 51/27 | 7/9 | 0.111 |
| SIS Participation, median | 80 | 63 | **0.002** |
| SIS Recovery, median | 84 | 63 | **0.003** |
| Life Satisfaction, n | | | |
| Satisfied/Not satisfied | 48/30 | 10/10 | 0.351 |
| **Fatigue problems, n (%)** | 57 (53) | 51 (47) | |
| Age, median | 66 | 61 | 0.060 |
| Sex, n | | | |
| Men/Women | 31/26 | 32/19 | 0.380 |
| Sense of coherence, median | 80 | 76 | 0.659 |
| Private financing, n | | | |
| Enough/Just enough or not enough | 37/20 | 23/28 | **0.040** |
| Stroke severity, n | | | |
| Mild/Moderate or Severe | 49/8 | 40/0 | 0.308 |
| Frenchay Activities Index, median | 32 | 30 | 0.864 |
| Involvement in care and treatment, n | | | |
| Involved/Involved to some extent or Not involved | 35/15 | 23/21 | 0.080 |
| SIS Participation, median | 88 | 90 | **0.004** |
| SIS Recovery, median | 90 | 89 | **0.036** |
| Life Satisfaction, n | | | |
| Satisfied/Not satisfied | 30/20 | 28/19 | 0.966 |
| **Emotional problems, n (%)** | 78 (71) | 32 (29) | |
| Age, median | 66 | 62 | **0.025** |
| Sex, n | | | |
| Men/Women | 46/32 | 18/14 | 0.793 |
| Sense of coherence, median | 81 | 70 | **0.007** |
| Private financing, n | | | |
| Enough/Just enough or not enough | 49/29 | 12/20 | **0.017** |
| SIS Emotion, median | 94 | 86 | 0.083 |
| Frenchay Activities Index, median | 32 | 27 | 0.256 |
| Involvement in care and treatment, n | | | |
| Involved/Involved to some extent or Not involved | 43/24 | 16/12 | 0.520 |
| SIS Participation, median | 88 | 66 | **0.003** |
| SIS Recovery, median | 80 | 70 | **0.003** |
| Life Satisfaction, n | | | |
| Satisfied/Not satisfied | 43/25 | 15/16 | 0.167 |
| **Concentration problems, n (%)** | 70 (65) | 37 (35) | |

(*Continued*)

**Table 1.** (Continued)

|  | Need met | Need met to some extent or Unmet* | *P*-value |
|---|---|---|---|
| Age, median | 68 | 59 | **0.003** |
| Sex, n |  |  |  |
| Men/Women | 43/27 | 19/18 | 0.316 |
| Sense of coherence, median | 81 | 70 | **0.007** |
| Private financing, n |  |  |  |
| Enough/Just enough or not enough | 45/25 | 14/23 | **0.010** |
| SIS Memory and thinking, median | 95 | 80 | **0.049** |
| Frenchay Activities Index, median | 31 | 30 | 0.580 |
| Involvement in care and treatment, n |  |  |  |
| Involved/Involved to some extent or Not involved | 44/18 | 14/18 | **0.012** |
| SIS Participation, median | 89 | 73 | **0.006** |
| SIS Recovery, median | 83 | 74 | **0.035** |
| Life Satisfaction, n |  |  |  |
| Satisfied/Not satisfied | 38/24 | 20/15 | 0.689 |
| **Memory problems, n (%)** | 76 (71) | 31 (29) |  |
| Age, median | 65 | 61 | 0.129 |
| Sex, n |  |  |  |
| Men/Women | 44/32 | 18/13 | 0.987 |
| Sense of coherence, median | 81 | 69 | **0.028** |
| Private financing, n |  |  |  |
| Enough/Just enough or not enough | 48/28 | 12/19 | **0.023** |
| SIS Memory and thinking, median | 95 | 80 | **0.050** |
| Frenchay Activities Index, median | 32 | 26 | 0.061 |
| Involvement in care and treatment, n |  |  |  |
| Involved/Involved to some extent or Not involved | 48/20 | 10/16 | **0.005** |
| SIS Participation, median | 84 | 72 | **0.017** |
| SIS Recovery, median | 80 | 68 | **0.042** |
| Life Satisfaction, n |  |  |  |
| Satisfied/Not satisfied | 42/25 | 15/15 | 0.243 |
| **Speaking difficulties, n (%)** | 85 (78) | 24 (22) |  |
| Age, median | 64 | 59 | 0.192 |
| Sex, n |  |  |  |
| Men/Women | 51/34 | 12/12 | 0.383 |
| Sense of coherence, median | 79 | 77 | 0.704 |
| Private financing, n |  |  |  |
| Enough/Just enough or not enough | 50/35 | 10/14 | 0.140 |
| SIS Communication, median | 94 | 81 | **0.003** |
| Frenchay Activities Index, median | 31 | 30 | 0.174 |
| Involvement in care and treatment, n |  |  |  |
| Involved/Involved to some extent or Not involved | 52/22 | 7/14 | **0.003** |
| SIS Participation, median | 84 | 63 | **0.005** |
| SIS Recovery, median | 80 | 63 | **0.004** |
| Life Satisfaction, n |  |  |  |
| Satisfied/Not satisfied | 48/29 | 10/12 | 0.160 |
| **Reading difficulties, n (%)** | 95 (88) | 13 (12) |  |
| Age, median | 64 | 65 | 0.445 |
| Sex, n |  |  |  |

(*Continued*)

**Table 1.** (Continued)

| | Need met | Need met to some extent or Unmet* | *P*-value |
|---|---|---|---|
| Men/Women | 57/38 | 6/7 | 0.347 |
| Sense of coherence, median | 80 | 70 | **0.032** |
| Private financing, n | | | |
| Enough/Just enough or not enough | 56/39 | 4/9 | 0.065 |
| Stroke severity, n | | | |
| Mild/Moderate or Severe | 83/0 | 8/2 | **0.024** |
| Frenchay Activities Index, median | 32 | 27 | 0.127 |
| Involvement in care and treatment, n | | | |
| Involved/Involved to some extent or Not involved | 55/31 | 4/5 | 0.260 |
| SIS Participation, median | 84 | 58 | **0.003** |
| SIS Recovery, median | 80 | 60 | **0.046** |
| Life Satisfaction, n | | | |
| Satisfied/Not satisfied | 55/33 | 3/8 | **0.036** |
| **Sight difficulties, n (%)** | 88 (82) | 19 (18) | |
| Age, median | 64 | 62 | 0.645 |
| Sex, n | | | |
| Men/Women | 52/36 | 10/9 | 0.606 |
| Sense of coherence, median | 81 | 71 | 0.151 |
| Private financing, n | | | |
| Enough/Just enough or not enough | 50/38 | 10/9 | 0.739 |
| Stroke severity, n | | | |
| Mild/Moderate or Severe | 75/13 | 15/4 | 0.500 |
| Frenchay Activities Index, median | 32 | 28 | 0.655 |
| Involvement in care and treatment, n | | | |
| Involved/Involved to some extent or Not involved | 50/28 | 8/8 | 0.294 |
| SIS Participation, median | 84 | 66 | 0.053 |
| SIS Recovery, median | 80 | 70 | 0.391 |
| Life Satisfaction, n | | | |
| Satisfied/Not satisfied | 50/31 | 8/9 | 0.267 |

*Need met to some extent: Mobility problems (20%), Falls (14%), Incontinence problems (12%), Pain (15%), Fatigue problems (19%), Emotional problems (14%), Concentration problems (9%), Memory problems (6%), Speaking difficulties (8%), Reading difficulties (4%), Sight difficulties (7%).

Findings in the present study indicate that having been involved in care and treatment can contribute to reporting met rehabilitation services needs related to mobility, falls and speaking. Involvement in discussions, planning and decisions on care and treatment are core components in person-centred care [53], patient participation [54] and shared decision-making [55, 56]. Shared decision-making aims to support patients' self-determination and autonomy by providing information and supporting deliberation. It has been suggested that defined steps in shared decision-making involve choice talk, option talk and decision talk together with the use of decision tools [57]. There is no information on the extent to which the participants in the present study were involved in such shared decision-making nor on healthcare services that can be defined as person-centred. However, in previous studies both patients [58] and staff [59] have described the informal parts of the patient-staff relationship, such as human connectedness and incorporating the patients' experiential knowledge in daily rehabilitation

**Table 2. Final logistic regression models for the association of the independent variables and met needs with regard to the 11 problem areas, odds ratios (OR) and 95% confidence intervals (CI).**

| *Problem area* Independent variables | Variable categorization | Odds for met needs OR (95% CI) |
|---|---|---|
| *Mobility, n = 71* | | |
| Involvement in care and treatment | Involved | 8.11 (2.28–28.84) |
| | Involved to some extent/Not involved | 1 |
| SIS participation | Decreased impact | 1.05 (1.01–1.09) |
| SIS mobility | Decreased impact | 1.07 (1.01–1.14) |
| *Area under the receiver operating characteristic curve = 0.857* | | |
| *R-Square = 0.494* | | |
| *Falls, n = 71* | | |
| Involvement in care and treatment | Involved | 11.96 (2.24–63.78) |
| | Involved to some extent/Not involved | 1 |
| SIS participation | Decreased impact | 1.07 (1.03–1.12) |
| SIS mobility | Decreased impact | 1.08 (1.02–1.15) |
| *Area under the receiver operating characteristic curve (ROC) = 0.907* | | |
| *R-Square = 0.597* | | |
| *Incontinence, n = 71* | | |
| Sense of coherence (SOC) | Increased SOC | 1.10 (1.02–1.19) |
| Frenchay Activities Index (FAI) | Increased FAI | 1.18 (1.07–1.30) |
| *Area under the receiver operating characteristic curve (ROC) = 0.932* | | |
| *R-Square = 0.557* | | |
| *Pain, n = 71* | | |
| Age | Increased age | 1.05 (1.00–1.11) |
| SIS participation | Decreased impact | 1.06 (1.02–1.10) |
| *Area under the receiver operating characteristic curve (ROC) = 0.802* | | |
| *R-Square = 0.280* | | |
| *Fatigue, n = 70* | | |
| Age | Increased age | 1.04 (1.00–1.08) |
| SIS participation | Decreased impact | 1.03 (1.01–1.06) |
| *Area under the receiver operating characteristic curve (ROC) = 0.714* | | |
| *R-Square = 0.161* | | |
| *Emotional, n = 71* | | |
| Sense of coherence (SOC) | Increased SOC | 1.07 (1.02–1.12) |
| SIS recovery | Increased recovery | 1.04 (1.01–1.07) |
| *Area under the receiver operating characteristic curve (ROC) = 0.801* | | |
| *R-Square = 0.354* | | |
| *Concentration, n = 70* | | |
| Age | Increased age | 1.05 (1.01–1.10) |
| SIS participation | Decreased impact | 1.05 (1.02–1.09) |
| *Area under the receiver operating characteristic curve (ROC) = 0.783* | | |
| *R-Square = 0.291* | | |
| *Memory, n = 70* | | |
| SIS participation | Decreased impact | 1.03 (1.00–1.06) |
| *Area under the receiver operating characteristic curve (ROC) = 0.652* | | |
| *R-Square = 0.100* | | |
| *Speaking, n = 71* | | |
| Involvement in care and treatment | Involved | 10.28 (2.24–47.30) |
| | Involved to some extent/Not involved | 1 |

*(Continued)*

**Table 2.** (Continued)

| *Problem area* Independent variables | Variable categorization | Odds for met needs OR (95% CI) |
|---|---|---|
| SIS communication | Decreased impact | 1.03 (1.00–1.07) |
| *Area under the receiver operating characteristic curve (ROC) = 0.767* | | |
| *R-Square = 0.323* | | |
| ***Reading, n = 71*** | | |
| Stroke severity | Mild | 15.25 (2.30–101.27) |
| | Moderate/Severe | 1 |
| *Area under the receiver operating characteristic curve (ROC) = 0.719* | | |
| *R-Square = 0.224* | | |
| ***Sight, n = 70*** | | |
| SIS participation | Decreased impact | 1.04 (1.00–1.07) |
| *Area under the receiver operating characteristic curve = (ROC) 0.712* | | |
| *R-Square = 0.117* | | |

sessions, as more important than formal decision-making and care planning to achieve experiences of involvement. Furthermore, it has been proposed that there are temporal aspects to consider as well. A review has suggested that patient-centred goal setting might not be suitable for all stages of rehabilitation and for all patients. Instead the process of goal setting needs to be tailored to individual patients' needs and preferences, which may change with time [28]. Nevertheless, experiences of involvement is a factor that most likely can be influenced by creating conditions that safeguard and support both formal and informal involvement in care and rehabilitation [56–58]. Considering the large proportion of people with stroke experiencing long-term unmet needs for health services [11–16] and the findings in this study, there is an urgent need to safeguard and support the patients' formal as well as informal involvement in care and rehabilitation.

The main strengths of the study are the long-term follow-up; the fact that all stroke patients admitted to Karolinska University Hospital's stroke units were eligible for inclusion in the original LAS-1 study; the use of face-to-face interviews for data collection which made it possible to include participants with various disabilities; and valid and reliable outcome measures covering both personal, disease specific and patient-reported outcomes. The mean age at stroke onset of the sample in the current 6-year follow-up was lower and a larger proportion had a mild stroke, in comparison with participants from the original LAS-1 study who were deceased or non-responders. The 48% deceased participants in our sample is comparable to results from large register based studies in Sweden, thus our sample could be considered representative for the general stroke population six years after stroke [60]. A limitation of the study might be the sample size, and that all variables with a p-value > 0.10 in the univariable analysis were excluded from the model regardless of their potential clinical significance, which may have limited the opportunity to discover other factors associated with met needs for rehabilitation. Further, we can not rule out that the participants' perceived needs may have been influenced by other independent variables than those included in the analyses.

## Conclusion

In a long-term perspective, there were several modifiable factors associated with met needs for rehabilitation services. The most prominent were perceived involvement in care and treatment, and perceived participation. These factors were more important for met rehabilitation services needs than disease specific factors six years after stroke.

## Acknowledgments

We would like to thank The UK Stroke Association for providing the Stroke Survivor Needs Survey Questionnaire.

## Author Contributions

**Conceptualization:** Charlotte Ytterberg, Hanne Kaae Kristensen, Malin Tistad, Lena von Koch.

**Formal analysis:** Charlotte Ytterberg.

**Funding acquisition:** Charlotte Ytterberg, Lena von Koch.

**Investigation:** Charlotte Ytterberg, Malin Tistad, Lena von Koch.

**Methodology:** Charlotte Ytterberg, Lena von Koch.

**Project administration:** Charlotte Ytterberg, Lena von Koch.

**Visualization:** Charlotte Ytterberg, Hanne Kaae Kristensen, Malin Tistad, Lena von Koch.

**Writing – original draft:** Charlotte Ytterberg, Hanne Kaae Kristensen, Malin Tistad, Lena von Koch.

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
