## [Decision Letter · Decision Letter 0]

2 Oct 2019

PONE-D-19-19035

Factors related to met needs for rehabilitation 6 years after stroke

PLOS ONE

Dear Dr. Ytterberg,

Thank you for submitting your manuscript to PLOS ONE. After careful consideration, we feel that it has merit but does not fully meet PLOS ONE’s publication criteria as it currently stands. Therefore, we invite you to submit a revised version of the manuscript that addresses the points raised during the review process.

We would appreciate receiving your revised manuscript by Nov 16 2019 11:59PM. To enhance the reproducibility of your results, we recommend that if applicable you deposit your laboratory protocols in protocols.io, where a protocol can be assigned its own identifier (DOI) such that it can be cited independently in the future. For instructions see: http://journals.plos.org/plosone/s/submission-guidelines#loc-laboratory-protocols

We look forward to receiving your revised manuscript.

Kind regards,

Janhavi Ajit Vaingankar

Academic Editor

PLOS ONE

Journal Requirements:

Additional Editor Comments (if provided):

Reviewers' comments:

Reviewer's Responses to Questions

**Comments to the Author**

1. Is the manuscript technically sound, and do the data support the conclusions?

Reviewer #1: Partly

Reviewer #2: Yes

Reviewer #3: Partly

Reviewer #4: Yes

2. Has the statistical analysis been performed appropriately and rigorously? 

Reviewer #1: Yes

Reviewer #2: Yes

Reviewer #3: No

Reviewer #4: No

3. Have the authors made all data underlying the findings in their manuscript fully available?

Reviewer #1: Yes

Reviewer #2: Yes

Reviewer #3: Yes

Reviewer #4: Yes

4. Is the manuscript presented in an intelligible fashion and written in standard English?

Reviewer #1: Yes

Reviewer #2: Yes

Reviewer #3: Yes

Reviewer #4: Yes

5. Review Comments to the Author

Reviewer #1: This is an important follow up study in rehabilitation after stroke. The authors describe that 6 years after stroke up to 47% of patients have unmet needs regarding fatigue or mobility.

This mono-centric and descriptive follow up of a previousrehabilitations study in stroke brings light into long termoutcome of stroke rehabilitation. Structure and language of the the manuscript are very well. However, before publication in a journal there are a few points that need to be adressed.

1. The study population is only around a third of the original study, and in regard of stroke onset there is a difference of almost 10 years. Still the study would benefit from somecomparison to the earlier data, i.e. to the data of all theincluded individuals in the first study.

2. Authors combine «need met to some extend» with «unmetneeds» which results in a bias to a negative outcome. Authorseither should use three groups: fully, partly, none met or clearly define to what percentage patients reporterd partlyunmet needs, counting as unmet.

3. Table 1 is rather loaded with data (6.5 pages) ; find a wayfor a better way of presenting the data.

Minor:

1. Title one is misspelled (« Factors related to meet needs …) anyway reviewer would prefer short title.

2. another limitation is the monocentric and local study design which therefore prevents a generalisability of the findings.

Reviewer #2: It is very researched and written manuscript which can be definitely considered for publication. My congratulations and appreciations to the authors.

Reviewer #3: Factors related to met needs for rehabilitation 6 years after stroke" (PONE-D-19-19035).

Reviewer comments

The paper by Ytterberg et al. reports on the association between met needs for rehabilitation 6 years after stroke and predefine factors. These factors included perceived needs for rehabilitation, personal factors, disease specific factors, and patient-reported disability factors. The paper is well written. Information are presented succinctly and clearly. The authors’ main conclusions are that the majority of participants reported having needs met at six years after stroke and that the most prominent factors associated with rehabilitation needs six years after the index stroke were perceived involvement in care and treatment, and perceived participation, which the authors pointed out were more important than disease specific factors. I invite the authors to consider the following suggestions which could improve the quality of the manuscript.

Minor points:

In the abstract, rather than saying that “the majority reported…” the authors may want to provide actual figures on met needs. This seems to be the bulk of their findings.

Also in the abstract, the authors conclude that “these factors were more important for having rehabilitation services needs met than disease specific factors six years after stroke”. This statement does not appear to be supported by the results provided in the summary.

I would also suggest that the introduction be shortened and to the point.

In the methods section, it would be useful to provide details about the validation (internal and external) of the questionnaire used in the study.

Specific difficulties inherent to administering the questionnaire to patient with disability from stroke (e.g. aphasia) should also be mentioned as well as approaches to mitigate them.

Giving the number and length of the questionnaires, it would be useful to specify how many sessions were needed to complete them as this has implications on the quality of answers provided.

Major points:

In the statistical analysis, it appears that all variables with a p-value > 0.10 in the univariate analysis were excluded from the model regardless of their potential clinical significance? For example, the variable sex was not included in the model but it is well known that stroke outcome is worse in women. This approach may have missed important predictors.

Given the large number of non-respondents, I suggest to compare the clinical and demographic characteristics of participants with those of the patients who declined to respond or could not be traced with the study participants. This would inform the reader about efforts from the authors to check for a potential non-respondents bias.

The authors report that stroke severity, which appears to be the only clinical predictor used in this analysis, was recorded using the Barthel Index. Stroke severity is often measured using the National Institute of Health stroke scale (NIHSS). The Barthel Index measures patient’s activity of daily living. In the context of the current study, the Barthel would ideally have been measured at 6-year follow up.

Reviewer #4: The study was conducted to explore the perceived needs for rehabilitation services of people six years after stroke and factors associated with having rehabilitation services needs met using data from a 6-year follow up of a prospective study on the rehabilitation process after stroke. This is interesting study and has advantage in terms long-term follow-up with face to face interviews in the participant's home by clinically experienced RA. However, i have few comments that need further clarification from the authors.

1. Under measurements and statistical analyses sections - authors mention that all measurements data were collected at the six-year follow-up except for data on stroke severity which were collected at stroke onset, and data on SOC which were collected at 12 months post stroke. The analyses were conducted using logistic regression models. Since this is a prospective study with 6 years follow up where certain variables were measured at different time point (i.e stroke severity, SOC), it is not clear how authors deals with effect of different time age of onset up to 6 years follow up on the outcomes variables as well as time-varying and invarying independent variables in their analysis. There is possibility that those with early or late onset stroke or longer or shorter duration of post stroke might have different needs for rehabilitation? In terms of time-varying and invarying effect from independent variable, there is possibility that stroke severity at stroke onset and level of SOC at 12-month post stroke might varied over time and at 6 months follow-up. I am not sure whether authors have captured those information in their study and have taken into account these effects in their analysis.

2. In current analysis, stepwise selection method followed by entered method was implemented in logistic regression models. There is bias and shortcoming when using stepwise approach in regression analysis which has been reported in literature (i.e Whittingham et al 2006. Why do we still use stepwise modelling in ecology and behaviour?J Anim Ecol. 2006 Sep;75(5):1182-9). Hence, it would be good if authors can clarify why authors decided using stepwise method plus entered method in their analysis instead of entered method alone!

In stepwise forward selection method, according to the authors variables with p =<0.05 were entered and P=> 0.10 were removed. It is not clear how variables with p >0.05 and p <0.10 were treated in the subsequent analysis?

Is there any significant interaction terms between predictors were found in current study?

There is no information about missing data and how it was handled in current analysis. It would be if authors can elaborate more about missing data and how missing data was treated in current analysis.

In Results section, authors mention that 44 declined and 18 could not be traced. I was wondering if authors can determine whether those who declined and could not be traced are different from those who remained in the study in terms of their baseline characteristics. Usually this information is very useful in order to ensure that the current results are not bias and can be generalize to the whole population cohort of the study.

6. PLOS authors have the option to publish the peer review history of their article (what does this mean?). If published, this will include your full peer review and any attached files.

Reviewer #1: No

Reviewer #2: Yes: Sureshkumar Kamalakannan

Reviewer #3: Yes: Alain Lekoubou

Reviewer #4: No

---

## [Author Response · Author response to Decision Letter 0]

28 Oct 2019

October 22, 2019

Dear Dr Vaingankar, Academic Editor at PLOS ONE 

Thank you for the invitation to submit a revised version of our manuscript PONE-D-19-19035 entitled “Factors related to met needs for rehabilitation 6 years after stroke”. We would like to thank the reviewers for their comments which we believe have been very helpful to improve the manuscript. Below are our responses to the reviewers’ comments and alterations made in the manuscript, marked yellow.

We would also like to add the funder “the Promobilia Foundation” in our financial disclosure. 

Please do not hesitate to contact me for any additional requests regarding this submission.

Sincerely,

Charlotte Ytterberg, corresponding author

Reviewer #1: This is an important follow up study in rehabilitation after stroke. The authors describe that 6 years after stroke up to 47% of patients have unmet needs regarding fatigue or mobility.

This mono-centric and descriptive follow up of a previous rehabilitation study in stroke brings light into long term outcome of stroke rehabilitation. Structure and language of the manuscript are very well. However, before publication in a journal there are a few points that need to be addressed. Authors’ response:

1. The study population is only around a third of the original study, and in regard of stroke onset there is a difference of almost 10 years. Still the study would benefit from some comparison to the earlier data, i.e. to the data of all the included individuals in the first study. In addition to the presented comparisons of sex and age, comparison of stroke severity between the study population and the original study have been added. Further, comparisons between the study population and those from the original study who were deceased or non-responders have been added. Results, first paragraph:

Their mean age at stroke onset was 63 years, ranging from 24 to 85 years, 64 (58%) were men, and 91 (83%) had a mild stroke severity. The mean age of all 349 participants in the original study group at stroke onset was 72 years, ranging from 24 to 95 years, 188 (54%) were men, and 213 (61%) had a mild stroke severity. The mean age at stroke onset of the 239 participants from the original study group who were deceased or non-responders was 76 years, ranging from 37 to 95 years, 125 (52%) were men, and 123 (51%) had a mild stroke severity.

2. Authors combine «need met to some extend» with «unmet needs» which results in a bias to a negative outcome. Authors either should use three groups: fully, partly, none met or clearly define to what percentage patients reported partly unmet needs, counting as unmet. Proportions of participants with needs met to some extent have been added in a footnote in Table 1:

*Need met to some extent: Mobility problems (20%), Falls (14%), Incontinence problems (12%), Pain (15%), Fatigue problems (19%), Emotional problems (14%), Concentration problems (9%), Memory problems (6%), Speaking difficulties (8%), Reading difficulties (4%), Sight difficulties (7%). 

3. Table 1 is rather loaded with data (6.5 pages); find a way for a better way of presenting the data. We agree that Table 1 is very long. One alternative would be to present only statistically significant variables for each problem area, however this would limit the interpretability. Another alternative would be to present 11 separate tables, one for each problem area. However, we are not certain that this would improve the readability. We would be grateful for any suggestion from the editor.

However, we believe the title of Table 1 was too lengthy, it has now been shortened:

Table 1. Characteristics of participants and univariable analyses of met and unmet needs by

problem area.

1. Title one is misspelled (« Factors related to meet needs …) anyway reviewer would prefer short title. We believe the title is spelled correctly. It refers to needs that have been met i.e., met needs.

2. another limitation is the monocentric and local study design which therefore prevents a generalisability of the findings. In the original LAS-I study, patients from the three stroke units at Karolinska University Hospital were eligible for inclusion i.e., more than one centre. However, to enable interpretation of generalisability of the findings we have added the following text in Discussion, last paragraph:

The 48% deceased participants in our sample is comparable to results from large register based studies in Sweden, thus our sample could be considered representative for the general stroke population six years after stroke (60).

Reviewer #2: It is very researched and written manuscript which can be definitely considered for publication. My congratulations and appreciations to the authors. 

Reviewer #3: The paper by Ytterberg et al. reports on the association between met needs for rehabilitation 6 years after stroke and predefine factors. These factors included perceived needs for rehabilitation, personal factors, disease specific factors, and patient-reported disability factors. The paper is well written. Information are presented succinctly and clearly. The authors’ main conclusions are that the majority of participants reported having needs met at six years after stroke and that the most prominent factors associated with rehabilitation needs six years after the index stroke were perceived involvement in care and treatment, and perceived participation, which the authors pointed out were more important than disease specific factors. I invite the authors to consider the following suggestions which could improve the quality of the manuscript. Authors’ response:

Minor points:

In the abstract, rather than saying that “the majority reported…” the authors may want to provide actual figures on met needs. This seems to be the bulk of their findings. The following has been added in the result section of the abstract:

In all problem areas the majority (53-88%) reported having needs met at six years after stroke, however 47% reported unmet needs regarding fatigue and 45% regarding mobility.

Also in the abstract, the authors conclude that “these factors were more important for having rehabilitation services needs met than disease specific factors six years after stroke”. This statement does not appear to be supported by the results provided in the summary. We have clarified this statement in the conclusion section of the abstract:

These factors had a stronger association with having rehabilitation services needs met than disease specific factors six years after stroke.

I would also suggest that the introduction be shortened and to the point. We agree with that and introduction should be short and to the point. However, we believe that length of the introdution is reasonable (1½ pages) to introduce the topic for readers not fully informed about the topic of the manuscript.

In the methods section, it would be useful to provide details about the validation (internal and external) of the questionnaire used in the study.

Specific difficulties inherent to administering the questionnaire to patient with disability from stroke (e.g. aphasia) should also be mentioned as well as approaches to mitigate them. The following information has been added in Methods/Measurements:

The SSNSQ was developed to assess perceived needs after stroke and included questions from validated questionnaires. It was validated in a review process by the King's College London Stroke Research Patients and Family Group (a service user research advisory group) (13).

And in the discussion, last paragraph: 

the use of face-to-face interviews for data collection which made it possible to include participants with various disabilities

Giving the number and length of the questionnaires, it would be useful to specify how many sessions were needed to complete them as this has implications on the quality of answers provided. The following has been added in Methods/Participants and procedures:

After written informed consent had been obtained, data were collected in one session through face-to-face interviews

Major points:

In the statistical analysis, it appears that all variables with a p-value > 0.10 in the univariate analysis were excluded from the model regardless of their potential clinical significance? For example, the variable sex was not included in the model but it is well known that stroke outcome is worse in women. This approach may have missed important predictors. We agree with the reviewer that this may be seen as a limitation and have added the following in the last paragraph of the discussion:

A limitation of the study might be the sample size, and that all variables with a p-value > 0.10 in the univariable analysis were excluded from the model regardless of their potential clinical significance, which may have limited the opportunity to discover other factors associated with met needs for rehabilitation. Further, the participants’ perceived needs may have been influenced by major incidents during the 6-year period, not related to the initial stroke incident.

Given the large number of non-respondents, I suggest to compare the clinical and demographic characteristics of participants with those of the patients who declined to respond or could not be traced with the study participants. This would inform the reader about efforts from the authors to check for a potential non-respondents bias. The following information has been added in the results, first paragraph:

Their mean age at stroke onset was 63 years, ranging from 24 to 85 years, 64 (58%) were men, and 91 (83%) had a mild stroke severity. The mean age of all 349 participants in the original study group at stroke onset was 72 years, ranging from 24 to 95 years, 188 (54%) were men, and 213 (61%) had a mild stroke severity. The mean age at stroke onset of the 239 participants from the original study group who were deceased or non-responders was 76 years, ranging from 37 to 95 years, 125 (52%) were men, and 123 (51%) had a mild stroke severity.

And in Discussion, last paragraph:

The 48% deceased participants in our sample is comparable to results from large register based studies in Sweden, thus our sample could be considered representative for the general stroke population six years after stroke (60).

The authors report that stroke severity, which appears to be the only clinical predictor used in this analysis, was recorded using the Barthel Index. Stroke severity is often measured using the National Institute of Health stroke scale (NIHSS). The Barthel Index measures patient’s activity of daily living. In the context of the current study, the Barthel would ideally have been measured at 6-year follow up. The following information has been added in Methods/Measurements/Independent variables, second paragraph:

Using the Barthel Index (35), which has shown good agreement with other stroke severity measures, stroke severity was categorized as recommended: mild (scores 50–100), moderate (scores 15–49) or severe (scores ≤14) (36).

Reviewer #4: The study was conducted to explore the perceived needs for rehabilitation services of people six years after stroke and factors associated with having rehabilitation services needs met using data from a 6-year follow up of a prospective study on the rehabilitation process after stroke. This is interesting study and has advantage in terms long-term follow-up with face to face interviews in the participant's home by clinically experienced RA. However, i have few comments that need further clarification from the authors. Authors’ response:

1. Under measurements and statistical analyses sections - authors mention that all measurements data were collected at the six-year follow-up except for data on stroke severity which were collected at stroke onset, and data on SOC which were collected at 12 months post stroke. The analyses were conducted using logistic regression models. Since this is a prospective study with 6 years follow up where certain variables were measured at different time point (i.e stroke severity, SOC), it is not clear how authors deals with effect of different time age of onset up to 6 years follow up on the outcomes variables as well as time-varying and invarying independent variables in their analysis. There is possibility that those with early or late onset stroke or longer or shorter duration of post stroke might have different needs for rehabilitation? In terms of time-varying and invarying effect from independent variable, there is possibility that stroke severity at stroke onset and level of SOC at 12-month post stroke might varied over time and at 6 months follow-up. I am not sure whether authors have captured those information in their study and have taken into account these effects in their analysis. All participants were followed up six years after stroke onset. However, we agree with the reviewer that the participants may have experienced incidents during the 6-year period that had an impact on their perceived needs. Thus, the following text has been added in the discussion, last paragraph:

Further, we can not rule out that the participants’ perceived needs may have been influenced by other independent variables than those included in the analyses.

The different time points for data-collection have been clarified in the last paragraph of Measurements:

All data were collected at the six-year follow-up except for data on stroke severity and age which were collected at stroke onset, and data on SOC which were collected at 12 months post stroke.

Further, information about SOC has been added in Methods/Measurements/Independent variables, second paragraph:

An adult individual’s SOC is considered to be relatively stable over time (34).

2.In current analysis, stepwise selection method followed by entered method was implemented in logistic regression models. There is bias and shortcoming when using stepwise approach in regression analysis which has been reported in literature (i.e Whittingham et al 2006. Why do we still use stepwise modelling in ecology and behaviour?J Anim Ecol. 2006 Sep;75(5):1182-9). Hence, it would be good if authors can clarify why authors decided using stepwise method plus entered method in their analysis instead of entered method alone!

 The following information has been added to clarify how missing data was handled and to motivate the use of stepwise and entered method:

A stepwise forward selection was used where variables with p≤0.05 were entered and those with p≥0.10 were removed. The Enter method was then used to verify a final model with more patients since several variables with missing data may have been excluded.

In stepwise forward selection method, according to the authors variables with p =<0.05 were entered and P=> 0.10 were removed. It is not clear how variables with p >0.05 and p <0.10 were treated in the subsequent analysis? In the stepwise forward selection procedure variables with p >0.05 and p <0.10 remain in the model as long as they are not p=> 0.10. 

Is there any significant interaction terms between predictors were found in current study? Interactions between predictors were not analyzed as we had no hypotheses about them.

There is no information about missing data and how it was handled in current analysis. It would be if authors can elaborate more about missing data and how missing data was treated in current analysis. The following information has been added in Analyses:

Participants with missing data in a model were excluded from that particular model.

In Results section, authors mention that 44 declined and 18 could not be traced. I was wondering if authors can determine whether those who declined and could not be traced are different from those who remained in the study in terms of their baseline characteristics. Usually this information is very useful in order to ensure that the current results are not bias and can be generalize to the whole population cohort of the study. In addition to the presented comparisons of sex and age, comparison of stroke severity between the study population and the original study have been added. Further, comparisons between the study population and those from the original study who were deceased or non-responders have been added. Results, first paragraph:

Their mean age at stroke onset was 63 years, ranging from 24 to 85 years, 64 (58%) were men, and 91 (83%) had a mild stroke severity. The mean age of all 349 participants in the original study group at stroke onset was 72 years, ranging from 24 to 95 years, 188 (54%) were men, and ??? had a mild stroke severity. The mean age at stroke onset of the 239 participants from the original study group who were deceased or non-responders was 76 years, ranging from 37 to 95 years, 125 (52%) were men, and 123 (51%) had a mild stroke severity.

And in Discussion, last paragraph:

The 48% deceased participants in our sample is comparable to results from large register based studies in Sweden, thus our sample could be considered representative for the general stroke population six years after stroke (60).

---

## [Decision Letter · Decision Letter 1]

19 Nov 2019

PONE-D-19-19035R1

Factors related to met needs for rehabilitation 6 years after stroke

PLOS ONE

Dear Dr. Ytterberg,

Thank you for submitting your manuscript to PLOS ONE. After careful consideration, we feel that it has merit but does not fully meet PLOS ONE’s publication criteria as it currently stands. Therefore, we invite you to submit a revised version of the manuscript that addresses the points raised during the review process.

One of the reviewers has concerns regarding sample size estimation and residual selection bias that may limit generalizability of the findings. I recommend that authors address these appropriately or alternatively, list these as some of the limitations of the work.

We would appreciate receiving your revised manuscript by Jan 03 2020 11:59PM. To enhance the reproducibility of your results, we recommend that if applicable you deposit your laboratory protocols in protocols.io, where a protocol can be assigned its own identifier (DOI) such that it can be cited independently in the future. For instructions see: http://journals.plos.org/plosone/s/submission-guidelines#loc-laboratory-protocols

We look forward to receiving your revised manuscript.

Kind regards,

Janhavi Ajit Vaingankar

Academic Editor

PLOS ONE

Reviewers' comments:

Reviewer's Responses to Questions

**Comments to the Author**

1. If the authors have adequately addressed your comments raised in a previous round of review and you feel that this manuscript is now acceptable for publication, you may indicate that here to bypass the “Comments to the Author” section, enter your conflict of interest statement in the “Confidential to Editor” section, and submit your "Accept" recommendation.

Reviewer #1: All comments have been addressed

Reviewer #3: (No Response)

Reviewer #4: All comments have been addressed

2. Is the manuscript technically sound, and do the data support the conclusions?

Reviewer #1: Yes

Reviewer #3: Yes

Reviewer #4: Yes

3. Has the statistical analysis been performed appropriately and rigorously? 

Reviewer #1: N/A

Reviewer #3: Yes

Reviewer #4: Yes

4. Have the authors made all data underlying the findings in their manuscript fully available?

Reviewer #1: Yes

Reviewer #3: No

Reviewer #4: No

5. Is the manuscript presented in an intelligible fashion and written in standard English?

Reviewer #1: Yes

Reviewer #3: Yes

Reviewer #4: Yes

6. Review Comments to the Author

Reviewer #1: Everything has been adressed adequately. The decision about table 1 should be with the publisher. Many thanks.

Reviewer #3: Overall, I was satisfied by the responses provided by the authors to my comments. However, I feel that some of the responses did not completely address my concerns.

- The authors state that a limitation of the study might be the sample size; this seems to contradict a prior assertion that their sample could be representative of the general stroke population in Sweden. I would be particularly cautious in referring to sample size as there is no indication in the manuscript how sample size was calculated.

- The authors have made significant efforts in providing demographic characteristics of the original sample; however it is still unclear if the non-respondents were different from the respondents. I encourage the authors to provide to perform a statistical analysis comparing at least the age and sex distribution of both groups (respondents and non-respondents)

- The authors state that the Barthel index is a measure of stroke severity. It is still my belief that the Barthel index measures disability and not stroke severity. Furthermore, measuring the Barthel index in the acute phase of stroke has significant pitfalls. The excellent review by Kasner ES (Lancet Neurol 2006; 5: 603–12) is a useful tool for understanding stroke scales. In that review, Kasner specifically point out that “In the setting of an acute stroke, the BI is not especially helpful as it is also highly susceptible to a “floor effect”. Most patients, even those with a minor stroke, are bedbound in the first few hours after stroke, either by their deficit or by medical directive, and therefore will initially receive low scores. Consequently, the BI cannot be used to measure initial stroke severity or, by extension, to stratify patients by severity in acute stroke trials”

Reviewer #4: Authors have addressed all my previous comments. I accept it as it is and I have no further comments.

7. PLOS authors have the option to publish the peer review history of their article (what does this mean?). If published, this will include your full peer review and any attached files.

Reviewer #1: No

Reviewer #3: Yes: Alain Lekoubou

Reviewer #4: No

---

## [Author Response · Author response to Decision Letter 1]

29 Nov 2019

Reviewer #3 

Overall, I was satisfied by the responses provided by the authors to my comments. However, I feel that some of the responses did not completely address my concerns.

- The authors state that a limitation of the study might be the sample size; this seems to contradict a prior assertion that their sample could be representative of the general stroke population in Sweden. I would be particularly cautious in referring to sample size as there is no indication in the manuscript how sample size was calculated. 

Authors' response

We agree that it may not be adequate to refer to sample size since this was an observational study and no sample size calculation was performed. We have therefore deleted sample size as a limitation.

Reviewer #3 

The authors have made significant efforts in providing demographic characteristics of the original sample; however it is still unclear if the non-respondents were different from the respondents. I encourage the authors to provide to perform a statistical analysis comparing at least the age and sex distribution of both groups (respondents and non-respondents)

Authors’ response

The following information has been added in the first paragraph of the results section:

There was no difference between study participants and non-responders in sex (p=0.203) but the non-responders were significantly older than the participants (p<0.001).

Reviewer #3 

The authors state that the Barthel index is a measure of stroke severity. It is still my belief that the Barthel index measures disability and not stroke severity. Furthermore, measuring the Barthel index in the acute phase of stroke has significant pitfalls. The excellent review by Kasner ES (Lancet Neurol 2006; 5: 603–12) is a useful tool for understanding stroke scales. In that review, Kasner specifically point out that “In the setting of an acute stroke, the BI is not especially helpful as it is also highly susceptible to a “floor effect”. Most patients, even those with a minor stroke, are bedbound in the first few hours after stroke, either by their deficit or by medical directive, and therefore will initially receive low scores. Consequently, the BI cannot be used to measure initial stroke severity or, by extension, to stratify patients by severity in acute stroke trials” 

Authors' response

Thank you for suggesting the review by Kasner ES which I have read with great interest. We agree that the Barthel Index is not useful within the first few hours after stroke when the patients are bedbound. However, the assessment with Barthel Index was in the present study performed within the first week. In addition, the BI was categorized in line with the recommendations in the article by Govan, Langhorne and Weir (Stroke 2009;40:3396-9): “BI, mRS, and SSS all have excellent agreement with each other when categorized” 

Clarification has been made in the manuscript, last paragraph of the methods section:

All data were collected at the six-year follow-up except for data on stroke severity and age which were collected within the first week after stroke onset, and data on SOC which were collected at 12 months post stroke.

---

## [Decision Letter · Decision Letter 2]

2 Jan 2020

Factors related to met needs for rehabilitation 6 years after stroke

PONE-D-19-19035R2

Dear Dr. Ytterberg,

We are pleased to inform you that your manuscript has been judged scientifically suitable for publication and will be formally accepted for publication once it complies with all outstanding technical requirements.

With kind regards,

Janhavi Ajit Vaingankar

Academic Editor

PLOS ONE

Additional Editor Comments (optional):

Reviewers' comments:

Reviewer's Responses to Questions

**Comments to the Author**

1. If the authors have adequately addressed your comments raised in a previous round of review and you feel that this manuscript is now acceptable for publication, you may indicate that here to bypass the “Comments to the Author” section, enter your conflict of interest statement in the “Confidential to Editor” section, and submit your "Accept" recommendation.

Reviewer #3: All comments have been addressed

2. Is the manuscript technically sound, and do the data support the conclusions?

Reviewer #3: (No Response)

3. Has the statistical analysis been performed appropriately and rigorously? 

Reviewer #3: (No Response)

4. Have the authors made all data underlying the findings in their manuscript fully available?

Reviewer #3: (No Response)

5. Is the manuscript presented in an intelligible fashion and written in standard English?

Reviewer #3: (No Response)

6. Review Comments to the Author

Reviewer #3: (No Response)

7. PLOS authors have the option to publish the peer review history of their article (what does this mean?). If published, this will include your full peer review and any attached files.

Reviewer #3: Yes: Alain Lekoubou

---

## [Editor Report · Acceptance letter]

3 Jan 2020

PONE-D-19-19035R2 

Factors related to met needs for rehabilitation 6 years after stroke 

Dear Dr. Ytterberg:

I am pleased to inform you that your manuscript has been deemed suitable for publication in PLOS ONE. Congratulations! Your manuscript is now with our production department. 

With kind regards,

on behalf of

Ms Janhavi Ajit Vaingankar 

Academic Editor

PLOS ONE